# The role of oxygen intake and liver enzyme on the dynamics of damaged hepatocytes: Implications to ischaemic liver injury via a mathematical model

Aditi Ghosh[1]☯*, Claire Onsager[1]☯, Andrew Mason[1]☯, Leon Arriola[1]‡, William Lee[2]‡, Anuj Mubayi[3,4]‡

**1** Department of Mathematics, University of Wisconsin - Whitewater, Whitewater, WI, United States of America, **2** Department of Hepatology, University of Texas Southwestern Medical Center, Dallas, TX, United States of America, **3** PRECESIONheor, Los Angeles, CA, United States of America, **4** Department of Mathematics, Illinois State State University, Normal, IL, United States of America

☯ These authors contributed equally to this work.
‡ These authors also contributed equally to this work.
* ghosha@uww.edu

**Data Availability Statement:** All relevant data are within the manuscript and its Supporting information files.

## Abstract

Ischaemic Hepatitis (IH) or Hypoxic Hepatitis (HH) also known as centrilobular liver cell necrosis is an acute liver injury characterized by a rapid increase in serum aminotransferase. The liver injury typically results from different underlying medical conditions such as cardiac failure, respiratory failure and septic shock in which the liver becomes damaged due to deprivation of either blood or oxygen. IH is a potentially lethal condition that is often preventable if diagnosed timely. The role of mechanisms that cause IH is often not well understood, making it difficult to diagnose or accurately quantify the patterns of related biomarkers. In most patients, currently, the only way to determine a case of IH is to rule out all other possible conditions for liver injuries. A better understanding of the liver's response to IH is necessary to aid in its diagnosis, measurement, and improve outcomes. The goal of this study is to identify mechanisms that can alter associated biomarkers for reducing the density of damaged hepatocytes, and thus reduce the chances of IH. We develop a mathematical model capturing dynamics of hepatocytes in the liver through the rise and fall of associated liver enzymes aspartate transaminase (AST), alanine transaminase (ALT) and lactate dehydrogenase (LDH) related to the condition of IH. The model analysis provides a novel approach to predict the level of biomarkers given variations in the systemic oxygen in the body. Using IH patient data in the US, novel model parameters are described and then estimated for the first time to capture real-time dynamics of hepatocytes in the presence and absence of IH condition. The results may allow physicians to estimate the extent of liver damage in an IH patient based on their enzyme levels and receive faster treatment on a real-time basis.

**Funding:** The authors did not receive any salary for this work and received no specific funding for this work.

**Competing interests:** The authors also declare that no competing interests exist.

# 1 Introduction

Ischaemic Hepatitis (IH) is a critical liver injury due to centrilobular liver cell necrosis with a massive increase in serum aminotransferase [1, 2]. It accounts for the most frequent cause of liver injury with 10% of patients admitted to hospital intensive care units [3–5]. Cell necrosis in Ischaemic Hepatitis is generally due to low oxygen extraction by hepatocytes, hepatic blood perfusion, systemic arterial hypoxemia and venous congestion [3]. The cause of IH is due to an underlying condition such as shock, trauma, or surgery. This underlying condition could be sudden, as is the case with shock (cardiac, allergic, hypovolemic, or septic) or could be a chronic condition such as heart failure [6]. Heart failure damages the blood flow through the portal vein, which in turn increases the hepatic vein's blood flow. The damage to the veins occurs over time without causing any immediate injury to the liver. For example, heart failure can result in a condition called congestive hepatopathy. Over a long period of time, this increased blood flow to the hepatic vein causes congestion of the vein, eventually building to a point of failure resulting in IH [7–9]. For this reason, there are no chronic cases of IH, just chronic conditions that can cause it. In addition to the causes, there are also other various conditions that can increase the risk for IH because they weaken the blood flow systems to the liver. A few such conditions are kidney disease, heart disease, and liver disease [6, 10, 11].

The liver is supplied with two types of blood delivery to protect against oxygen impairment through two different veins namely the portal vein and the hepatic vein. The portal vein delivers two thirds of the liver's blood. Despite this, the portal vein only provides 50% of the oxygen needed. The other 50% is provided by the oxygen rich hepatic blood. In order to protect itself against damage, the liver has a process called the 'hepatic arterial buffer response' [1, 12]. Through this process, damage to the portal vein results in an increase in blood flow through the hepatic vein. Similarly, damage to the hepatic vein leads to an increase in the oxygen levels of the blood through the portal vein. This extra layer of protection implies that significant injury to one or both the portal vein system and arterial hepatic system is necessary to cause ischaemic hepatitis. The damage must be severe enough that it reduces the percentage of oxygen received in the liver [13–15]. Oxygen delivery (DO2) is the volume of oxygen delivered to the systemic vascular bed per minute. Systemic oxygen delivery ($DO2$) in the body depends on several factors like cardiac output, haemoglobin level and oxygen saturation of haemoglobin. Normal value for ($DO_2$) lies within 520 and 720 ml$O_2$ /min.$m^2$ and often low cardiac output can bring down ($DO_2$) to the critical threshold for hypoxia which is 330 ml$O_2$ /min.$m^2$ [1, 12, 16–19]. This work here focuses on developing a mathematical model to understand liver damage due to IH with respect to systemic oxygen delivery (DO2) in the body.

Once IH occurs, it is characterized by abnormal liver function, tissue damage, and tissue death that leads to liver failure. There is also a rapid rise in various serum levels such as aspartate transaminase (AST), alanine transaminase (ALT) and lactate dehydrogenase (LDH) [1, 20–22]. These elevated levels are often the first sign of liver injury. Once this is realized, physicians determine the type of liver injury that has occurred. Typically there is an abrupt rise of AST, ALT and LDH levels within 12 -24 hours after the initiating event. After the treatment of the underlying IH causing condition, the AST level decreases toward normal faster than the ALT level. An increased Serum lactate level is also an important biomarker towards the detection of IH [3].

The current diagnosis method for IH results from the exclusion of all other possible causes of liver damage. At that point, the patient is treated by correction of the underlying condition that is causing IH [6]. This long process of exclusion leaves many cases untreated before the damage is too extensive. As a result, only 50% of hospital patients diagnosed with IH survive [5, 21, 23, 24]. A better understanding of IH is needed to aid in diagnosis and treatment in

order to improve the odds. Medical research is mostly limited to the data collected once patients are treated and data collected through experiments on rats and other small animals. A mathematical model can provide a new perspective and understanding of the condition. In present work, we develop a mathematical model to study the dynamics of IH with respect to systemic oxygen delivery in the body and measure the biomarkers associated with it, as IH is marked by massive increases of serum aminotransferase.

There are currently no mathematical models for ischaemic liver injury or hypoxic hepatitis. There are however, models of other types of liver injury, models of liver cell death, and models of biomarker production that helped us better understand and build this model [25]. **The goals of this study are to: (i) provide a framework of relevant biomarkers and hepatocytes in order to study patterns of IH condition, (ii) estimate average decay and growth parameters related to concentration of biomarkers during IH, (iii) predict real time peak levels of different biomarkers, namely AST, ALT, and LDH, due to liver injury caused by IH at varied oxygen levels in the body as well as varied treatment frequency, and (iv) identify threshold conditions for time to reach the critical concentration level of hepatocytes for irreversible damage for varied initial oxygen level in the body**. This study's framework involves building a mathematical model and focus on the above goals. The model provides formulation of time to reach a state of 70% damaged hepatocyte (a critical condition of liver injury) starting from fixed oxygen level and from set initial state of liver under no treatment scenario. Once 70% of the liver's hepatocytes are damaged, the liver is unable to regenerate and a transplant may be necessary [26]. Knowing the time to reach a critical state of damaged hepatocyte will aid physicians to identify when there is an immediate need of liver transplant and allow a manipulation of drug treatment. This model also explores an estimate of level of further damage due to reperfusion. We validated our model results using data from [1, 5, 27, 28] that represents clinical conditions underlying hypoxic hepatitis.

We organize the paper as follows. Section 2 provides a detailed description of the mathematical model. Equilibrium points and stability analysis are discussed in Section 3. We also provide a sensitivity analysis of the equilibrium points with respect to an important parameter here. Numerical simulation of the mathematical model to predict the levels of bio-markers for a given value of delivered oxygen in the blood is presented in section 4. This model is also validated from data collected from literature in this section. We conclude with a discussion in Section 5.

## 2 Materials and methods

### 2.1 Data sources

We develop a functional form of changes in oxygen delivery in the body over time using empirical information from patient data in previously published articles which we have provided in details in S1 Data. In this study, we used data corresponding to 50% initial oxygen level (the most common oxygen level observed in cases of ischaemic injury). The parameters describing the variations in AST, ALT, LDH are estimated using [2, 21] (the definition of parameters are given in Table 1. The parameter describing the rate of oxygen return $\epsilon$ in Eq 7 is chosen based on the peak biomarker levels for 50% oxygen treated at 8 hours that closely matches the averages found in the literature [2, 21] (estimates of parameters are collected in Table 1.

In order to validate the model, a compiled data from four empirical studies on ischaemic liver injury [1, 5, 27, 28] are used and is provided in the S1 Data. The relevant data, needed for fitting model results, are extracted to include only cases of ischaemic hepatitis due to cardiac injury or reduced oxygen. For example, the data for low $O_2$ availability, low $O_2$ delivery, and

**Table 1. Table of variables.**

| Variables | Unit | Description | |
|---|---|---|---|
| $O$ | unitless | Amount of oxygen | |
| $A$ | *pmol* | Amount of total ATP in the liver | |
| $H$ | *cells* | Number of healthy hepatocytes in a liver | |
| $Z$ | *cells* | Number of damaged hepatocytes in a liver | |
| $S$ | IU/L | Amount of AST per liter of blood | |
| $L$ | IU/L | Amount of ALT per liter of blood | |
| $D$ | IU/L | Amount of LDH per liter of blood | |
| **Parameters** | **Unit** | **Description** | **Source** |
| $A_{norm}$ | $1.6 \times 10^9$ *pmol* | Amount of ATP in a healthy liver | [29] |
| $\rho$ | 1.43 *pmol/cell/day* | ATP production rate | [30] |
| $k$ | 1.43 *pmol/cell/day* | ATP consumption rate | [30] |
| $r$ | 1 *days*$^{-1}$ | Functional hepatocyte regeneration rate | [25] |
| $\eta$ | 6.381 *days*$^{-1}$ | Cell necrosis rate | [31] |
| $\delta_Z$ | 5 *days*$^{-1}$ | Damaged hepatocyte lysis rate | [25] |
| $H_{max}$ | $1.6 \times 10^{11}$ *cells* | Maximum number of hepatocytes possible in a healthy liver (assumed constant) | [25] |
| $\theta$ | 5 L | Total amount of blood in human body | [25] |
| $\delta_S$ | 0.92 *days*$^{-1}$ | AST clearance rate | [25] |
| $\delta_L$ | 0.35 *days*$^{-1}$ | ALT clearance rate | [25] |
| $\delta_D$ | 0.459 *days*$^{-1}$ | LDH clearance rate | [32] |
| $\beta_S$ | 20000 *IU* | Total amount of AST in a healthy liver | [2, 21] |
| $\beta_L$ | 9000 *IU* | Total amount of ALT in a healthy liver | [2, 21] |
| $\beta_D$ | 200000 *IU* | LDH production rate | [1] |
| $S_{min}$ | 12 *IU/L* | Minimum AST level | [25] |
| $L_{min}$ | 9 *IU/L* | Minimum ALT level | [25] |
| $D_{min}$ | 120 *IU/L* | Minimum LDH level | [33] |
| $D_{max}$ | 30000 *IU/L* | Maximum LDH level | [2] |
| $O_0$ | varied | Oxygen fraction from cardiac injury | Estm. |
| $\epsilon$ | 20 *days*$^{-}$1 | Rate of oxygen return | [34] |
| $\tau$ | *days* | Time of treatment | Estm. |

low $O_2$ are collected from [27]. Similarly, from [1], only the data for acute cardiac failure and congestive heart failure are used for comparison. Since most biomarker data is collected at unknown intervals after ischaemic injury, the peak values are the best way to compare data. From the four papers [1, 5, 27, 28], an average peak serum value for AST ranging from 1927 − 4587 IU/L, ALT ranging from 1803 − 1959 IU/L, and LDH ranging from 3067 − 4494 IU/L. Different data are used for parameter estimation and model validation. Model parameters are estimated using patient data from studies [2, 21].

Validation of the model is carried out by comparing the model outputs with the corresponding quantities from data (such as peak biomarker values observed in hospitalized cases of IH due to cardiac injury or reduced oxygen; data obtained from [1, 5, 27, 28]). These quantities are compared because currently, it is an effective clinical way to verify patient conditions, since the blood serum levels are standard checks for hospital patients. For AST, we have a maximum of 2681.95 IU/L with 50% oxygen availability in the body treated after 8 hours and around 3000 IU/L after 10 hrs which fits in the range 1927 − 4587 IU/L. For ALT, we have a maximum level of 1316.89 IU/L with 50% oxygen availability in the body treated after 8 hours and 1500 IU/L after 10 hours which falls short of the range 1803 − 1959 IU/L, and LDH with

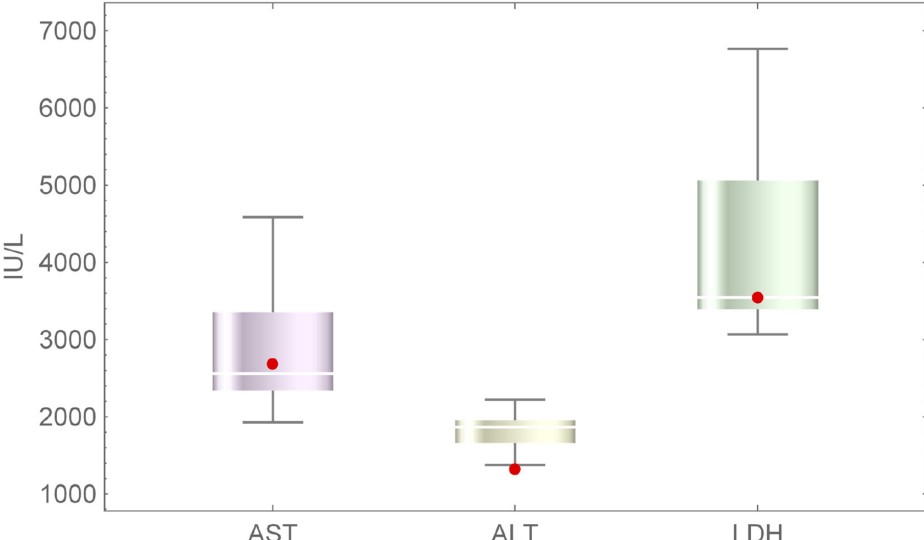

**Fig 1. Bio-marker data collected from literature to include cases of ischaemic hepatitis due to low $O_2$ availability, low $O_2$ delivery.** The (mean, standard deviation) IU/L of AST, ALT, LDH are (2877.0, $\sqrt{868.2}$), (1821.4, $\sqrt{266.6}$), (4273.8, $\sqrt{1486.9}$) respectively.

50% oxygen availability in the body treated after 8 hours has a maximum of 3547.93 IU/L and around 3900 IU/L treated after 10 hours which fits in the range $3067 - 4494$ IU/L in the literature (see Fig 1). Most of these values are well within the overall ranges listed in the literature. The larger literature averages specially for ALT could possibly be due to data taken from more patients treated much later than 8 hours or starting with less than 50% oxygen. Other error could have come from a difference in the number of patients whose data are analyzed to gather the averages in each paper. The data set used for validation of the model underlying the results as well as for parameter estimation is provided in details in multiple sheets in an excel document as S1 Data. All the tables from previously published articles used for validation are also provided in the S1 Data.

## 2.2 Model description

We model hepatocyte death over time and focus on cell death due to depletion of adenosine triphosphate ATP(A is used as the symbol for ATP in the model) in the liver cells. Cells use energy source from ATP molecules and die due to dearth of it as it lacks the energy to carry out essential functions. This form of cell death is called necrosis (see [35–37]). We only consider necrosis as a form of cell death in our model based on consistent understanding in the literature to study ischaemia/hypoxia caused by a decrease in oxygenated blood flow due to heart failure [2, 5, 21].

According to [2], ischaemic hepatitis was originally defined in 1979 as "liver injury characterized by a centrilobular liver cell necrosis with a sharp increase in serum aminotransferase activity in the setting of cardiac failure." The process of cytokine storm as a contributing factor to ischaemic hepatitis resulting from heart failure is not considered much in literature. While it is an important process, it is not related to the focus of this work. We do not model liver injury due to immune responses and cytokine release, but rather due to a lack of oxygen and insufficient ATP. In the literature, there is some disagreement on the role of apoptosis as a cell death mechanism in this instance. While many sources, such as those mentioned above, define

liver ischaemia solely in terms of necrosis, others suggest that apoptosis may play a role. For example, [1] mentions a debate on the type of cell death involved and that recent studies have reported evidence on the role of apoptotic cell death in addition to necrotic (oncotic) cell death. Despite this, the inclusion of apoptosis does not yet appear widely accepted within the literature. For instance, newer articles such as [13, 15] still characterize hypoxic hepatitis as a result of necrosis. For this reason, we choose to focus only on necrosis at this time and consider apoptosis for future studies red. To model the form of cell death due to necrosis we track the production and consumption of ATP. The majority of ATP in the human body is produced and consumed locally, as a result, our model assumes that all ATP production relevant to the liver is produced in the liver. Any ATP produced outside the liver is considered negligible here. Additionally, the majority of ATP production requires oxygen in a process called cellular respiration. This process provides 95% of the ATP in each cell [38, 39]. When there is a lack of oxygen being delivered to the liver, it fails to produce enough ATP, resulting necrosis [36].

Since the production of ATP requires oxygen, the amount of oxygen (O) the liver receives determines how much or how little ATP can be produced and thus how many cell death occurs [38]. Our model assumes a starting value of oxygen which is then restored at a certain time of treatment. We use a logistic function to better fit the physiology as the liver takes time to regain oxygenated blood [34]. **We assume here that if the initial oxygen present is 70% or less in the body, there are chances of liver injury due to Ischaemia. Also, if 70% of the hepatocyte is damaged due to lack of oxygen, the liver does not recover on its own and a liver transplant is needed for survival** [26]. **Since data corresponding to 50% initial oxygen level is mostly available in literature, we will focus our result based on 50% initial oxygen level in the body**.

We also model the production of aspartate aminotransferase-AST (S), alanine aminotransferase-ALT (L) and lactate dehydrogenase-LDH (D) which are all important bio-markers and byproducts of liver damage. The levels of these bio-markers can be observed via blood tests and are often used in determining the type or severity of liver injury [6]. AST and ALT behave in a similar fashion with both being produced at a certain rate based on damaged hepatocytes (see Fig 2). AST is found in most major organs throughout the body and is released when any of the cells are damaged [14]. ALT is only found in the liver making it a major indicator of liver injury [14]. As a result, AST levels in the blood rise much more than ALT in cases of cardiac injury, because the heart also releases AST. This means that the more liver damage that happens over a certain amount of time the higher the production of these enzymes will be.

Lactate dehydrogenase (LDH) is produced in cases of liver damage. However, it has been noticed to be much higher in liver injury due to a lack of oxygen. This is because LDH is produced as back up for use when cells are not getting enough ATP [40]. LDH is part of the ATP production process. This enzyme is a catalyst for the reaction that converts pyruvate, NADH, and Hydrogen into lactate and NAD+, and back. In a healthy liver, most of the ATP is produced through a combination of pyruvate and oxygen.

In cases of low oxygen, such as in IH, the body compensates by producing more through an anaerobic process called glycolysis. This process uses NAD+ to produce enough ATP to keep the liver functioning until oxygen is returned. The increased production of LDH speeds up the conversion of pyruvate and NADH into NAD+ and allows the production of a small amount of ATP [38]. This process results in an increase in lactate, also referred to as lactic acid. The resulting build up of lactate halts this production of LDH, allowing levels to normalize [40]. Since this process is only meant as a temporary fix, the increase in LDH occurs rapidly in a matter of hours before capping off at its maximum and slowly decreasing [1]. Furthermore, the resulting ATP produced is minimal compared to that produced by the remaining hepatocytes [38]. For this reason, we considered ATP produced by LDH to be negligible in this

Ischemic Liver Model

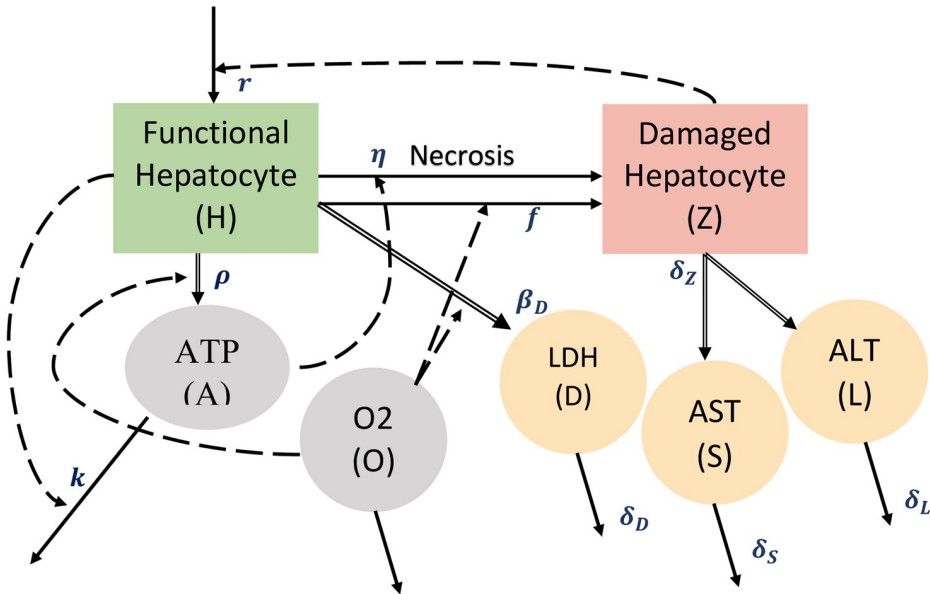

**Fig 2. Schematic depiction of the model.** *Solid arrow* indicates conversion of the same variable from one state to another, *dotted arrow* indicates influence of a variable on the transition rate, and *double arrow* indicates secretion of biomarkers. See expressions terms of the Model System 1–6 to understand the appropriateness of arrows. The initial conditions used to solve the ODE system are $H^* = H_{max}$, $A^* = A_{norm}$, $Z^* = 0$, $S^* = S_{min}$, $L^* = L_{min}$, $D^* = D_{min}$.

model. Fig 2 shows the schematic depiction of the model. Solid arrow indicates conversion of the same variable from one state to another, dotted arrow indicates influence of a variable on the transition rate, and double arrow indicates secretion of biomarkers. See expressions in terms of the model to understand the appropriateness of arrows.

$$\frac{dA}{dt} = \rho H O(t) - kH \frac{A}{A_{norm}} \tag{1}$$

$$\frac{dH}{dt} = rH\left(1 - \frac{H+Z}{H_{max}}\right) - \eta H\left(1 - \frac{A}{A_{norm}}\right) - f(O'(t)) \tag{2}$$

$$\frac{dZ}{dt} = \eta H\left(1 - \frac{A}{A_{norm}}\right) - \delta_Z Z + f(O'(t)) \tag{3}$$

$$\frac{dS}{dt} = \frac{\delta_Z \beta_S}{\theta H_{max}} Z - \delta_S(S - S_{min}) \tag{4}$$

$$\frac{dL}{dt} = \frac{\delta_Z \beta_L}{\theta H_{max}} Z - \delta_L(L - L_{min}) \tag{5}$$

$$\frac{dD}{dt} = \frac{H\beta_D}{\theta H_{max}}(1 - O(t))(1 - \frac{D}{D_{max}}) - \delta_D(D - D_{min}) \tag{6}$$

where

$$O(t) = \frac{1 - O_0}{1 + e^{-\epsilon(t-\tau)}} + O_0 \tag{7}$$

$$f(O') = \left| \frac{d}{dt} O(t) \right| = \left| \frac{\epsilon(1 - O_0)e^{-\epsilon(t-\tau)}}{\left(1 + e^{-\epsilon(t-\tau)}\right)^2} \right| = \begin{cases} \dfrac{\epsilon(1 - O_0)e^{-\epsilon(t-\tau)}}{\left(1 + e^{-\epsilon(t-\tau)}\right)^2} & if \ \ O_0 \leq 1 \\[3mm] \dfrac{\epsilon(O_0 - 1)e^{-\epsilon(t-\tau)}}{\left(1 + e^{-\epsilon(t-\tau)}\right)^2} & if \ \ O_0 > 1 \end{cases} \tag{8}$$

The term $f(O'(t))$ is the term for the damage due to reperfusion given by Eq 8. Note that this term is zero before treatment. Also, the right hand side of the system explicitly depend on time(t). The first term $rH\left(1 - \frac{H+Z}{H_{max}}\right)$ of the healthy hepatocyte model controls the production of new healthy hepatocytes. This term is scaled by the hepatocyte growth rate r and has a carrying capacity of that total number of hepatocytes. This implies that the regrowth of the liver is limited by the death of its cells. However, this condition is not permanent, as the damaged hepatocytes are removed from the liver over time making room for new cells to grow. The second term $\eta H\left(1 - \frac{A}{A_{norm}}\right)$ controls the death of healthy hepatocytes. This term is controlled by the current amount of ATP as opposed to its normal level and is scaled by the death rate $\eta$. The damaged hepatocytes are created using the death term from the healthy hepatocyte function. In this function the death term $\eta H\left(1 - \frac{A}{A_{norm}}\right)$ is added. This creates a one to one conversion from healthy hepatocytes to damaged hepatocytes. The second term $\delta_Z Z$ controls the rate at which damaged hepatocytes are flushed from the liver. It is based on the lysis rate $\delta_Z$. ATP is also broken into two terms: the production term $\rho HO$ and the consumption term $kH\frac{A}{A_{norm}}$. Both terms are based on the number of healthy hepatocytes (H), as they are both producing and consuming the ATP in the liver. The consumption is capped by $\frac{A}{A_{norm}}$ to keep ATP from going negative which would not make physiological sense. The production, however, is controlled by the percentage of oxygen (O). The production and consumption of ATP is controlled by the two rates $\rho$ and k. In our model we assume that these two rates are equal. For our oxygen we use a logistic function in order to model the return of oxygen to the liver after treatment.

The return of blood and oxygen to the liver following treatment is called reperfusion. Since the reperfusion of blood to the liver takes roughly 6 hours [34], we chose $\epsilon$ to be 20 here to match our peak biomarker levels for 50% oxygen treated at 8 hours that closely matches the averages from the literature [2, 21]. Additionally, we augment $\tau$ by a certain amount to offset the curve in order to have the time of treatment be reflected by the beginning of the rise in oxygen. With the logistic function we used, $\tau$ controls the midpoint of the curve so for all figures $\tau$ is offset by $\frac{3}{24}$ (3 hours). The biomarkers AST and ALT are modeled the same way as in [41, 42]. The first terms of these equations model the production of the biomarkers by the damaged hepatocytes Z. This production is controlled by the lysis rate of damaged hepatocytes $\delta_Z$, the amount of blood in the human body $\theta$, the maximum number of hepatocytes $H_{max}$ and the total amount of each biomarker in the liver $\beta_S$ and $\beta_L$. The second term is the removal of the biomarkers from the bloodstream at a rate given by $\delta_S$ or $\delta_L$ for AST and ALT respectively. The minimum observed value of the biomarkers are used to prevent the levels from ever dropping below their minimum. The LDH equation is similar to the other biomarkers, with growth depending on the maximum hepatocytes and the total amount of blood in the human body, with a growth rate of $\beta_D$ The first term differs from the other biomarkers in few ways where

AST and ALT are produced by damaged hepatocytes, LDH is produced by the remaining healthy hepatocytes to help the liver and as such we remove the dependence of the damaged hepatocyte lysis rate. We also include the term $1 - O$ to match the physiology [40]. This allows the LDH level to increase when oxygen is less than 1, i.e. when the liver is being damaged. We also include the term $\left(1 - \frac{D}{D_{max}}\right)$ to prevent the level of LDH from increasing beyond the maximum dictated by the physiology [40]. Similar to the other biomarkers, LDH is removed from the bloodstream at a rate $\delta_D$ and capped to not go below the minimum observed level $D_{min}$. Most of the parameters in this model are obtained from existing literature as shown in Table 1. However, some of the parameters in our model are estimated to achieve physiologically reasonable behaviour. We have used Python to solve our system of ODE's of our model and for parameter estimation. The initial conditions to solve the model are $H^* = H_{max}$, $A^* = A_{norm}$, $Z^* = 0$, $S^* = S_{min}$, $L^* = L_{min}$, $D^* = D_{min}$.

## 3 Results and discussion

### 3.1 Analysis (a case when $O(t)$ is independent of t, i.e., $O$ is constant)

The steady states of $A$, $H$, $Z$, $S$, $L$, and $D$ are solved assuming a full oxygen percentage, or $O = 1$ when is oxygen is independent of t as shown in Fig 3. The amount of oxygen, given in Eq 7 reaches saturation after treatment and does not affect the equilibrium analysis. This reveals different steady states based on the relation of the ATP production and consumption rates, $\rho$ and $k$. The trivial case is $H^* = 0$, $Z = H_{max}$, $A^* = 0$ and the bio-markers above the minimum which happens when a patient dies. We now consider the case when $H^*$ is nonzero. Here we create a new variable $\lambda$ such that $\lambda = \left(1 - \frac{\rho}{k}\right)$. There exist the following steady states:

$$
\begin{aligned}
H^* &= H_{max}\frac{1 - \frac{\eta}{r}\lambda}{1 + \frac{\eta}{\delta_Z}\lambda} \\
A^* &= A_{norm}(1 - \lambda) \\
Z^* &= \frac{\eta\lambda}{r}H_{max}\left[\frac{r - \eta\lambda}{\delta_Z + \eta\lambda}\right] \\
S^* &= \frac{\delta_Z\beta_S\eta\lambda}{r\delta_S\theta}\left[\frac{r - \eta\lambda}{\delta_Z + \eta\lambda}\right] + S_{min} \\
L^* &= \frac{\delta_Z\beta_L\eta\lambda}{r\delta_L\theta}\left[\frac{r - \eta\lambda}{\delta_Z + \eta\lambda}\right] + L_{min} \\
D^* &= D_{min}
\end{aligned}
\tag{9}
$$

These equations reveal restrictions on the possible values of $\lambda$. The ideal case is when $\lambda = 0$, which happens when $\rho = k$. In this case we get the following steady states from the above equations:

$$
\begin{aligned}
H^* &= H_{max} \\
A^* &= A_{norm} \\
Z^* &= 0 \\
S^* &= S_{min} \\
L^* &= L_{min} \\
D^* &= D_{min}
\end{aligned}
\tag{10}
$$

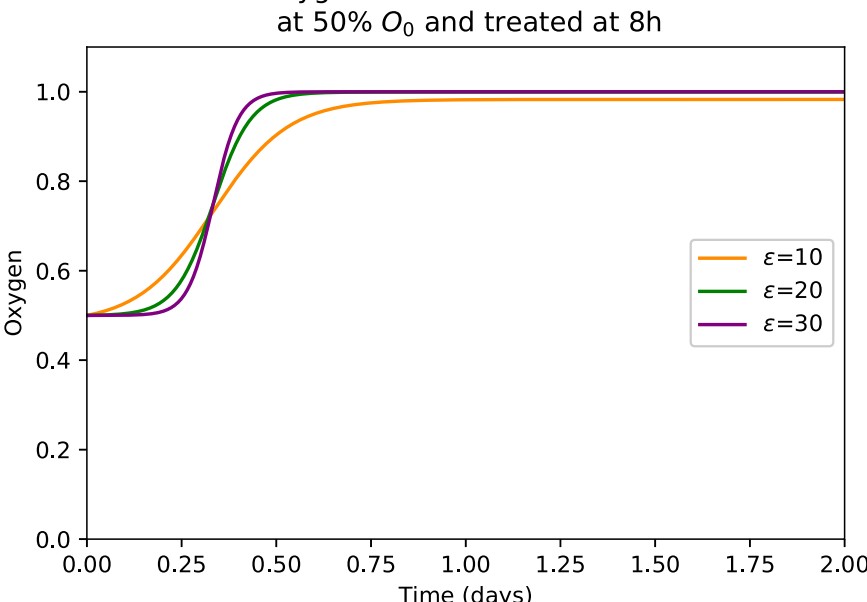

**Fig 3. Demonstration of the oxygen function at 50% initial $O_0$ values for different values of $\epsilon$.**

This case is for our model assumption. Since oxygen is set to be full in the steady state, the patient would be healthy or treated when this steady state occurs. A healthy patient have hepatocyte levels $H$ returned to a maximum, resulting in zero damaged hepatocytes $Z$. This would also result in normal levels of ATP and the biomarkers AST, ALT, and LDH which are the normal/minimum values listed above. It is for this reason that the value of $\rho$ is assumed to be equal to $k$ in our model.

In the cases where $\rho \neq k$ we run into steady states that are physically impossible. We examine two possibilities,

- Consider, $\lambda < 0$. When $\lambda$ is negative, as is the case where $\frac{\rho}{k} > 1$, a negative steady state for the damaged hepatocytes $Z^*$ can occur. This is the case where $\frac{\delta_Z}{\eta} > |\lambda|$. A negative $Z$ value corresponds to a negative amount of damaged hepatocytes and an amount of healthy hepatocytes greater than the amount of hepatocytes in the liver which cannot realistically occur. Similarly, if $\frac{\delta_Z}{\eta} < |\lambda|$ the steady state of the healthy hepatocytes is negative. This negative state would not only imply that the patient is dead, but also that they had more damaged hepatocytes than the total number of hepatocytes they started with. Analysis of the remaining case where $\frac{\delta_Z}{\eta} = |\lambda|$ revealed the trivial steady state where there are no healthy nor damaged hepatocytes present. The failure of all possible relations between $\lambda$ and $\frac{\delta_Z}{\eta}$ when $\lambda < 0$ prove that $\lambda$ must always be positive.

- Consider, $\lambda > 0$. The possibility for $\lambda$ to be positive also fails under certain conditions. Both the healthy and damaged hepatocytes would be negative if $\frac{r}{\eta} < \lambda$ and zero if $\frac{r}{\eta} = \lambda$.

Therefore, the only physically possible steady state that occurs when $\rho \neq k$ is that where $\frac{\rho}{k} < 1$ and $\frac{r}{\eta} > 1 - \frac{\rho}{k}$.

These resulting steady states however, do not return to normal values expected of a healthy patient. This indicates that there is still an underlying condition that is damaging the liver. Since we are assuming treatment at this point, this too is an unrealistic condition for our model. We determine the stability of the steady states by analyzing the Jacobian of the system of differential equation. Linearizing the system of intracellular and hepatocyte differential equations yields the Jacobian

$$
J = \begin{bmatrix}
-\frac{kH}{A_{norm}} & \rho O - \frac{kA}{A_{norm}} & 0 & 0 & 0 & 0 \\
\frac{\eta H}{A_{norm}} & r\left(1 - \frac{H+Z}{H_{max}}\right) - \frac{rH}{H_{max}} - \eta\left(1 - \frac{A}{A_{norm}}\right) & -\frac{rH}{H_{max}} & 0 & 0 & 0 \\
-\frac{\eta H}{A_{norm}} & \eta\left(1 - \frac{A^*}{A_{norm}}\right) & -\delta_Z & 0 & 0 & 0 \\
0 & 0 & \frac{\delta_Z \beta_S}{\theta H_{max}} & -\delta_S & 0 & 0 \\
0 & 0 & \frac{\delta_Z \beta_L}{\theta H_{max}} & 0 & -\delta_L & 0 \\
0 & \frac{\beta_D}{\theta H_{max}}(1-O)\left(1 - \frac{D}{D_{max}}\right) & 0 & 0 & 0 & -\frac{H\beta_D}{\theta H_{max}}\left(\frac{(1-O)}{D_{max}}\right) - \delta_D
\end{bmatrix}
$$

The system has unique steady state that is numerically seen to be locally stable. In particular, corresponding to the equilibrium points Eq 10 the eigenvalues corresponding to the Jacobian are $-\delta L$, $-r$, $-\delta S$, $-\delta Z$, $(-k * H_{max})/A norm$, $-\delta D$. This healthy equilibria characterized by Eq 10 is locally stable.

## 3.2 Variation in the equilibrium with respect to $\rho$ and $k$

We now discuss the sensitivity of the equilibrium points, in other words we determine how perturbations to the parameter $\rho$ and k affect the equilibrium points $H^*$, $Z^*$, $Z^*$, $S^*$ and $L^*$ [43]. Specifically we calculate the normal sensitivity index of $H^*$ with respect to $\rho$ [43] given by $SI[H^*; \rho] = -\frac{\eta k \rho (\delta_Z + r)}{(k(\delta_Z + \eta) - \eta \rho)(k(\eta - r) - \eta \rho)}$. Similarly we calculate the normal sensitivity index of the other variables as shown in [43] with respect to $\rho$ and k. The magnitude of sensitivity index outlines the relative importance of that variable with respect to equilibrium and the sign represents either proportional or reciprocal relationship. We plot the sensitivity index of these variables using Mathematica in Fig 4. As seen in Fig 4, increasing $k$ by 1% yields an approximate 3.3% increase in $Z^*$, $S^*$, $L^*$. All these three equilibrium points have almost equal impact on $k$. Similarly a 1.8% increase occurs in $H^*$. This implies that $Z^*$, $S^*$, $L^*$ have higher sensitivity compared to $H^*$. Also, $Z^*$, $S^*$, $L^*$ has negative sensitivity index with respect to $\rho$ which implies if $\rho$ decreases then the equilibrium point increases.

## 3.3 Numerical results (a case when $O(t)$ is explicit function of t)

We consider three scenarios here how biomarkers changes with time as treatment $\tau$, how biomarkers changes with initial $O_0$ level present in the body, how biomarkers changes with and without treatment and what is the rate of irreversible damage with changes in initial oxygen $O_0$. Here we have considered oxygen $O(t)$ as explicit functions of time t. We numerically solve the system of Eqs 1–6 using ode solver in Python. Results are generated by initializing a certain time of treatment and then varying the initial oxygen level. The lower the initial oxygen level, the more severe the damage to the liver will be.

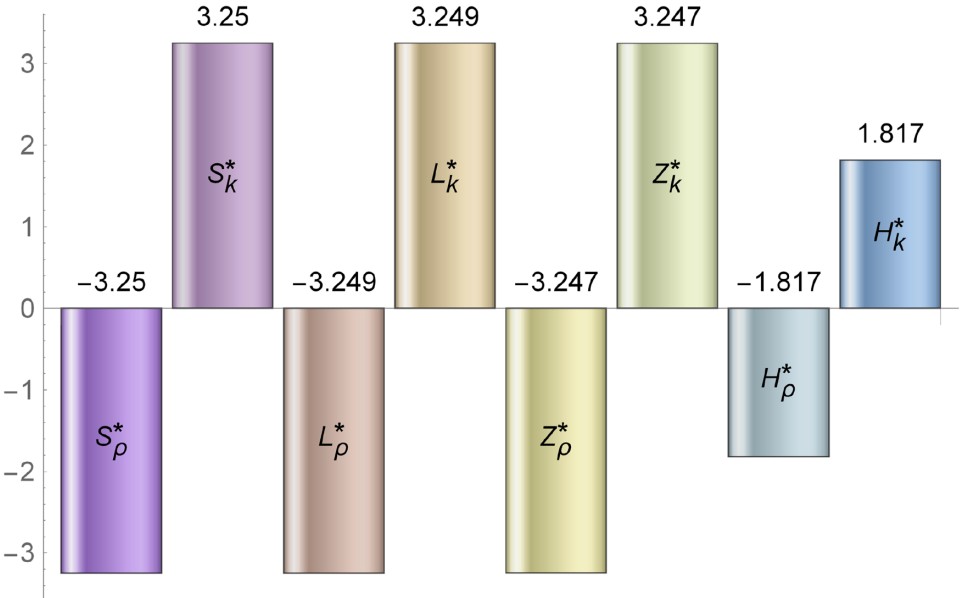

**Fig 4. Sensitivity indexes of the equilibrium concentration of variables with respect to parameters $\rho$ and $k$.** Note, $\rho$ negatively and $k$ positively impacts equilibrium concentration of all model variables.

The liver is able to regrow even after sustaining an incredible amount of damage, however there is a threshold from which a liver cannot regrow itself. This threshold is roughly around 30% of healthy hepatocytes [26] but it can vary slightly from individual to individual. Thus, any patient with liver damage over 70% may be at risk of liver failure and is most likely in need of a liver transplant [26]. This critical point is indicated by a dotted line on the graphs Fig 8 of hepatocytes. In any instances where the hepatocyte curve dips below this line, critical damage is sustained.

### 3.4 Hepatocyte damage at varied treatment frequency

Figs 5b, 6b, 7b and 8b shows the plot of ALT, AST, LDH and Hepatocyte values varied at different oxygen levels treated after eight hours. The increased severity of hepatocyte damage can also be observed through a sharper rise in AST, ALT and LDH. In this case AST spikes faster than ALT and also returns to its normal value more rapidly while ALT takes over a week to return to its normal value. These observed slope changes and normalization times appear to be similar to what is observed with experimental data [1]. LDH also spikes very quickly, however it gets capped off at an upper limit. This is because LDH production in the liver slows as LDH builds up eventually hitting a certain limit at which point the body ceases LDH production [40].

### 3.5 Hepatocyte damage at varied levels of initial oxygen

Plots at Figs 5a, 6a, 7a and 8a are generated by taking a common initial oxygen level in cases of IH and varying the time of treatment. The common level selected from the literature is 50% [1]. As expected, the longer the oxygen is left below its normal value, the more the liver is damaged. Since the oxygen level is the same across all treatment times, the damage happens at the same rate up until the time of treatment.

(a)

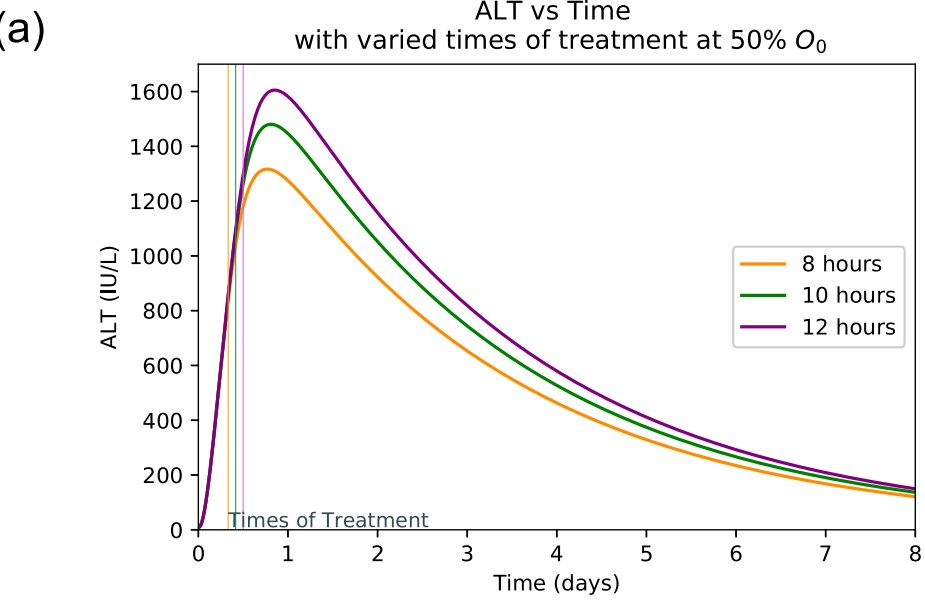

(b)

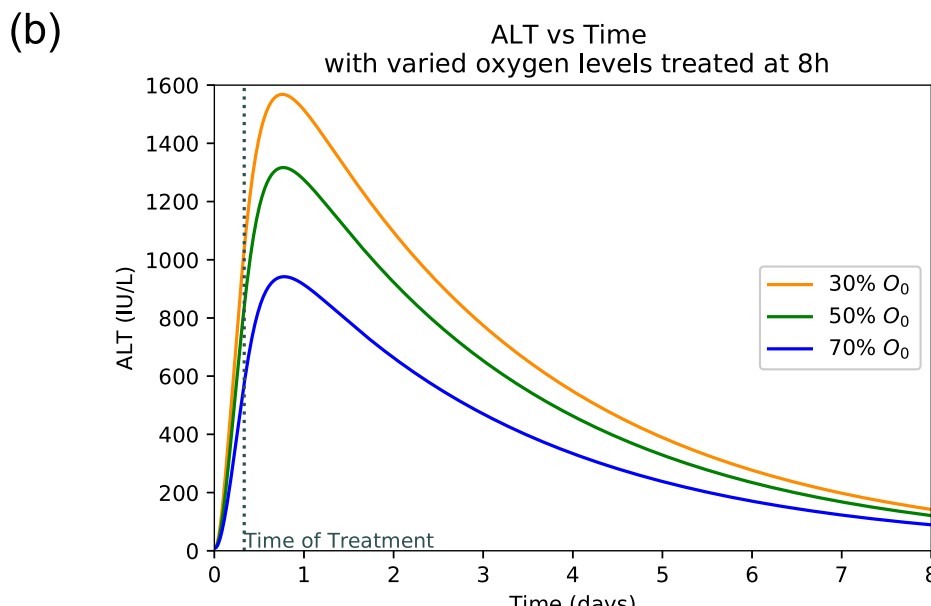

**Fig 5. ALT peaks with varied times of treatment and oxygen levels.** (**a**) ALT peaks with varied times of treatment at 50% oxygen level. (**b**) ALT peaks with varied initial oxygen levels treated after 8 hours.

## 3.6 Critical threshold of damaged hepatocytes for liver transplant

We observed in Fig 8b that at almost 40% oxygen the liver will reach critical damage if treated at or after 8 hours. If treated sooner, the patient will be less likely to need a liver transplant. Due to the return of oxygen over a period of 6 hrs, there is still damage for a while after treatment is administered [34]. This correlates to the space between the treatment and the time of the trough observed in this figure. The associated AST and ALT graphs show similar results

(a)

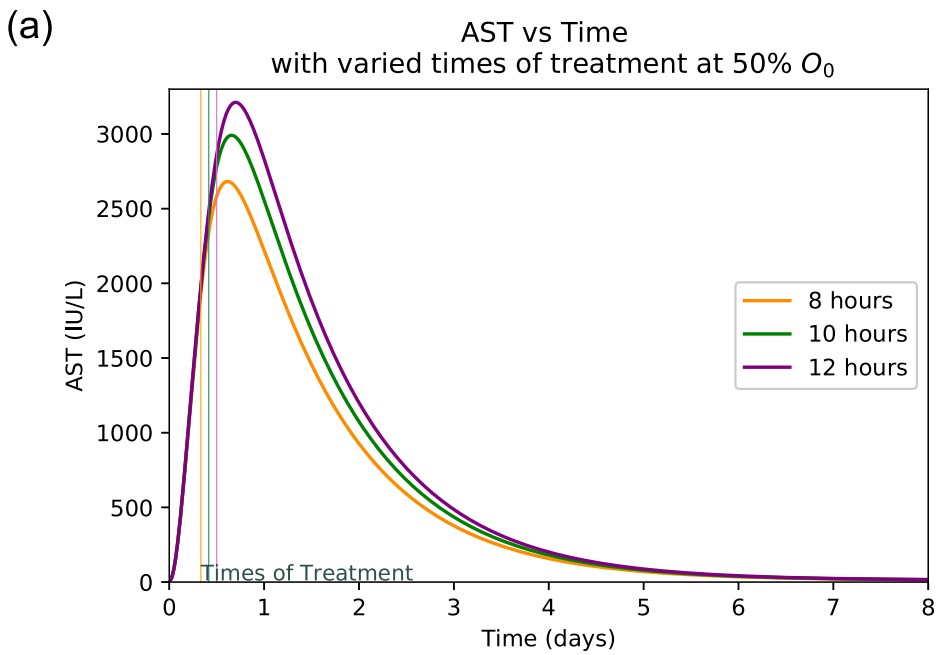

(b)

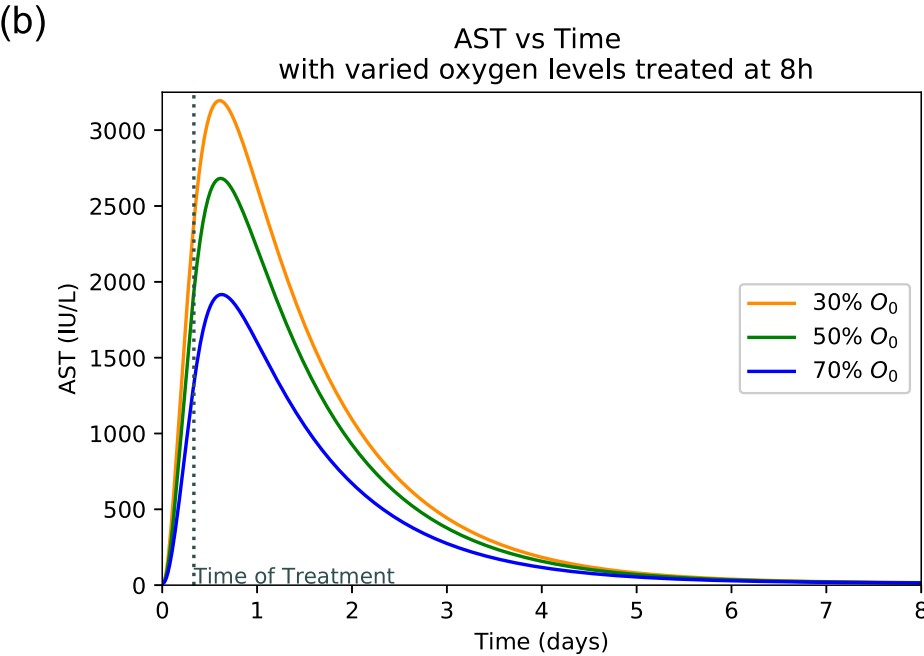

**Fig 6. AST peaks with varied times of treatment and oxygen levels.** (**a**) AST peaks with varied times of treatment at 50% oxygen level. (**b**) AST peaks with varied initial oxygen levels treated after 8 hours.

with the peaks also occurring at time after treatment and increasing with prolonged untreated time.

Interestingly, when there is no treatment given AST, ALT and LDH spikes and begins to decay, despite the fact that liver cell death is still occurring. This is because the number of damaged hepatocytes are removed through a process called lysis [25]. Since AST and ALT are

(a)

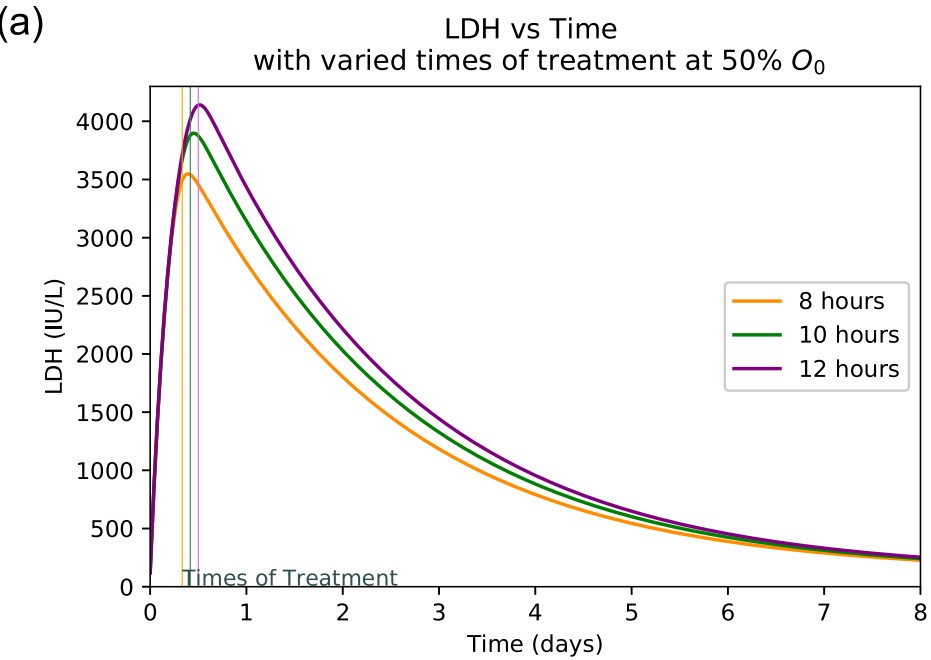

(b)

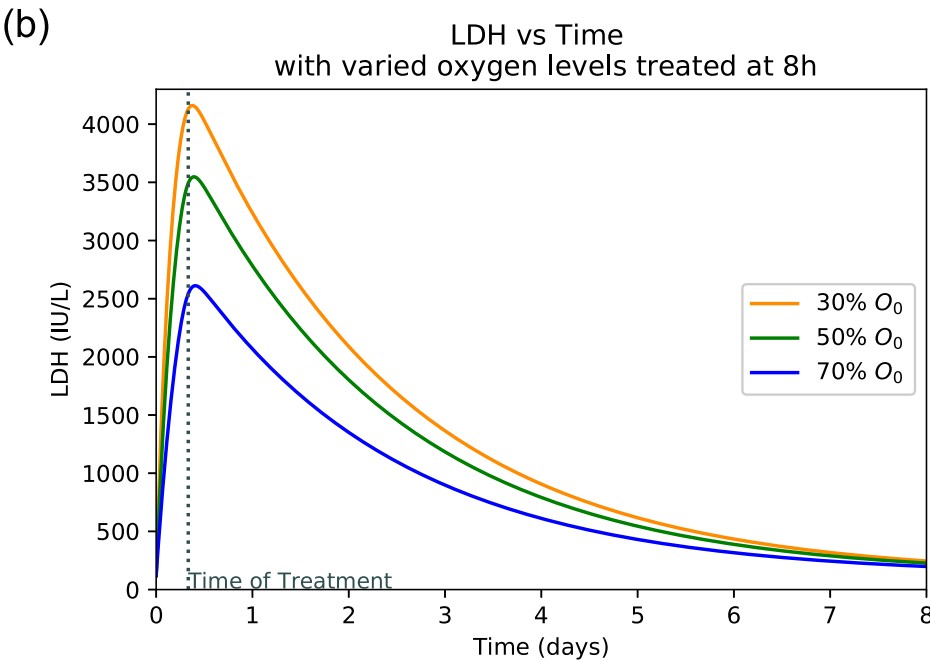

**Fig 7. LDH peaks with varied times of treatment and oxygen levels.** (**a**) LDH peaks with varied times of treatment at 50% oxygen level. (**b**) LDH peaks with varied initial oxygen levels treated after 8 hours.

directly related to the amount of damaged hepatocytes, this explains the decay while there being no treatment. LDH, on the other-hand, returns to normal due to the fact that it is produced by healthy hepatocytes [40]. With no treatment being administered the number of healthy hepatocytes eventually dwindles to zero and LDH, AST, ALT production ceases.

(a)

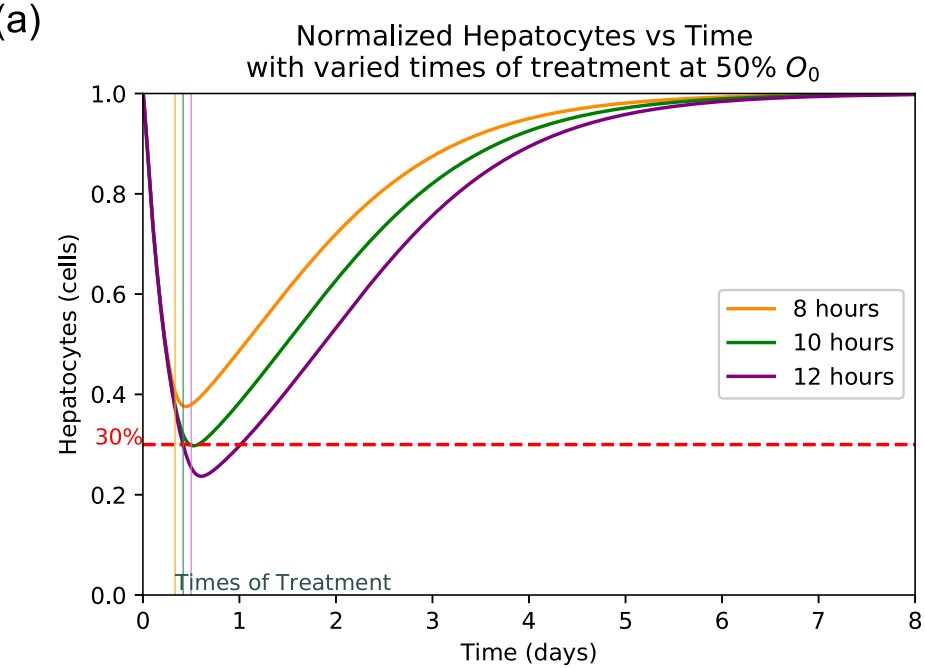

(b)

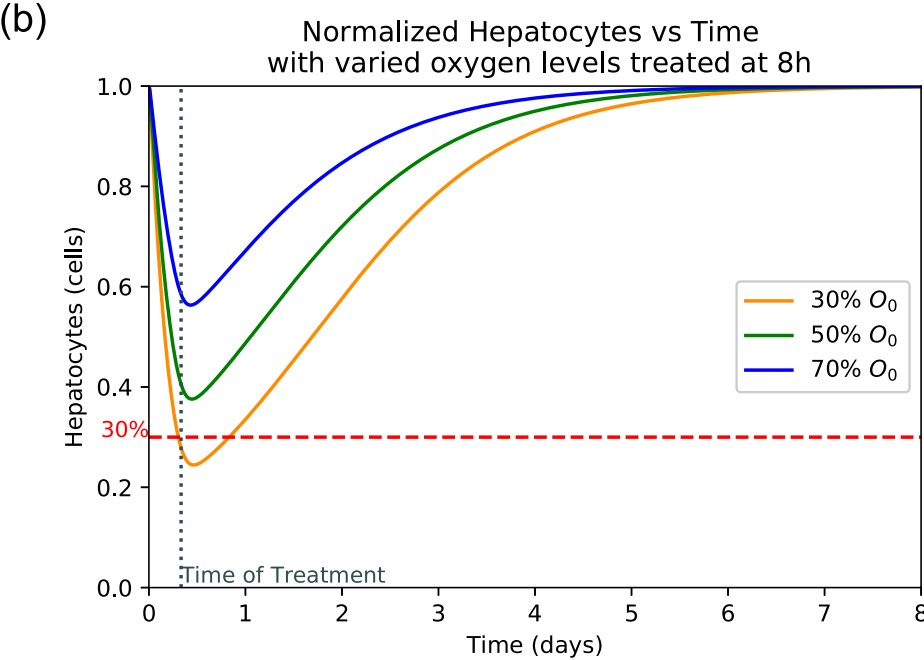

**Fig 8. Hepatocyte damage with varied times of treatment and oxygen levels.** (**a**) Healthy Hepatocytes as function of normal with varied times of treatment at 50% oxygen level. (**b**) Healthy Hepatocytes as a function of normal with varied initial oxygen levels treated after 8 hours.

Through Fig 9a our model shows different time scenarios of hepatocyte damage beyond repair for varied oxygen levels when there is no treatment. This means that with a lower initial oxygen level, the time before liver damage becomes severe (dropping below 30% remaining healthy hepatocytes) is shorter than if the initial oxygen level is higher. Fig 9b shows the

(a)

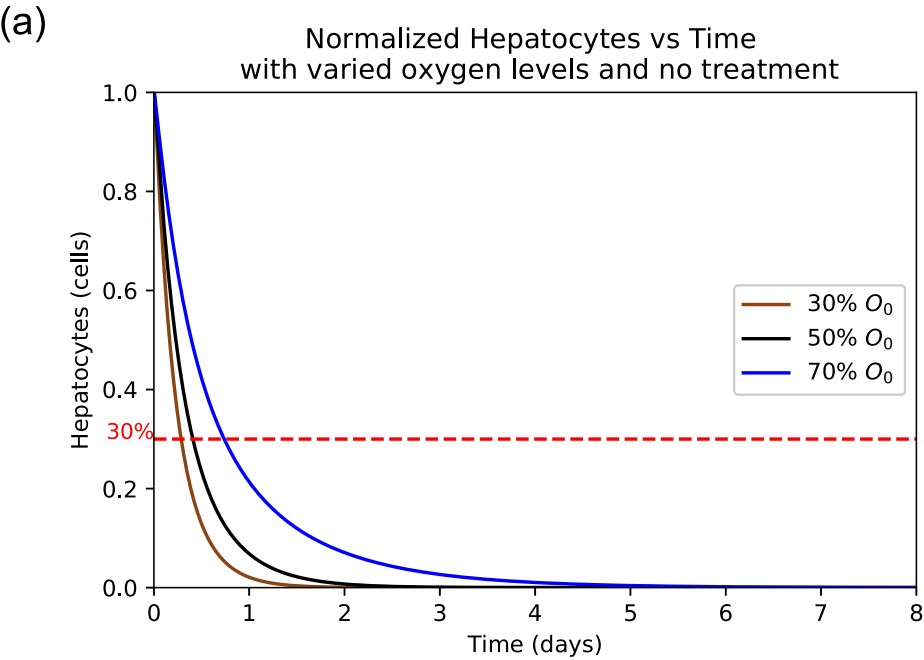

(b)

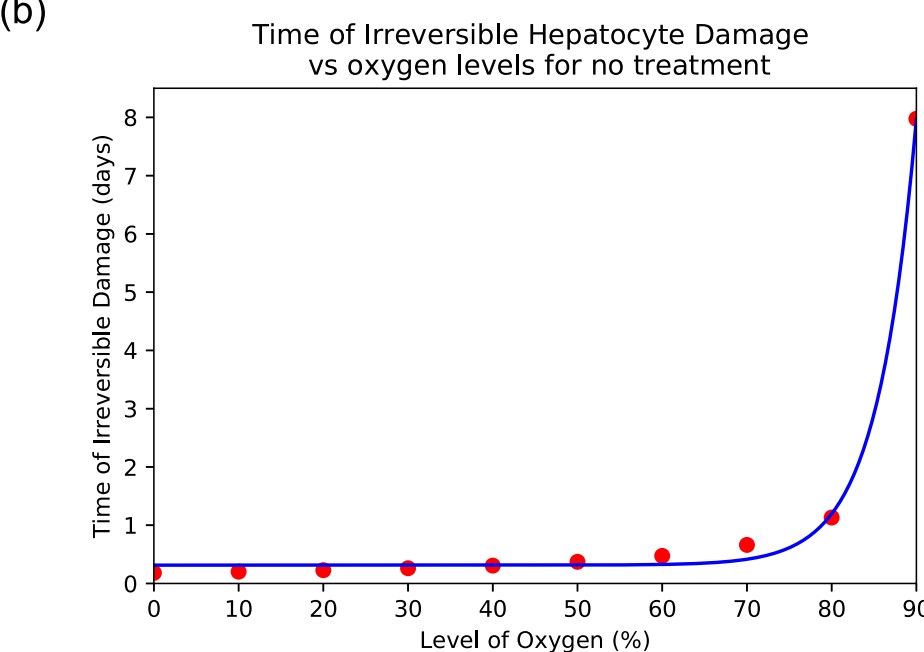

**Fig 9. Demonstration of the Hepatocyte, AST, ALT, LDH level with varied O2 levels when there is no treatment.** (**a**) Hepatocyte death with varied O2 levels when there is no treatment. (**b**) The estimated time of reaching the critical 30% Hepatocyte level for irreversible damage given the initial oxygen level.

estimated timing of reaching the critical 30% hepatocyte level for irreversible damage for different initial $O_2$ level. The scatter plot fits the exponent function $f(x) = ae^{kx} + c$, where $a = 2.68176(10^{-8})$, $k = 0.216333$ and $c = 0.314243$ are estimated using Mathematica. The model also helps estimate a time frame for treatment as shown in Fig 9a. Thus this plot may help

physician to estimate the time of irreversible damage given different levels of initial oxygen avaliability in patients. It is also observed in Fig 8a and 8b that hepatocytes begin to recover shortly after treatment is administered which indicates that this model can aid physician estimate the time for treatment.

### 3.7 Ischaemia-reperfusion injury

Ischaemia-reperfusion injury (IRI) or reperfusion injury or reoxygenation injury, is the tissue damage caused when blood supply returns to tissue after a period of ischaemia or lack of oxygen (anoxia or hypoxia). The absence of oxygen and nutrients from blood during the ischaemic period creates a condition in which the restoration of circulation results in inflammation which leads to the death of the cells and oxidative damage through the induction of oxidative stress rather than restoration of normal function [34]. There is a sizable amount of cell death that happens when the liver is reperfused with oxygenated blood [44]. Current sources disagree on how much of the damage a liver with IH sustains is from reperfusion and how much is from necrosis [1, 14, 34]. Because of this, we model damage due to reperfusion separately from our main model.

In order to model the damage due to reperfusion, we use the derivative of our oxygen function. In this case we have $f(O'(t)) = \left|\frac{dO}{dt}\right|$. This is because the damage during reperfusion is caused by the rapid influx of oxygenated blood [34]. By using the absolute value of the derivative, we take into account equal damage due to the rate of oxygen decrease and increase. This accounts for any damage that occurs due to the sudden loss of oxygen when IH occurs, as well as reperfusion damage since both cases cause the mass amount of inflammation leading to cell death [34]. Since this term is damage to the liver, it is taken away from healthy hepatocytes and added to damaged hepatocytes.

The absolute value is multiplied by a variable that we call the reperfusion rate c. With units of cells/day the value of c is the rate at which hepatocytes die due to reperfusion damage. Through our research of reperfusion injury, we are unable to determine this rate without further data or reconstruction of our oxygen equation. Reperfusion also creates organ damage following transplantation due to the return of oxygenated blood. As a result, much of the literature we obtain on the topic of reperfusion focuses on transplant cases where different organs are completely reperfused over periods of time longer than expected in our case [45, 46]. This leads us to estimate the values of c for the model. We assume that the damage due to reperfusion does not exceed that due to ischaemia. Therefore, we select values of c that produces less damage than our original model. Trial of various c values reveals that values larger than c = 2 produces damage comparable to our ischaemia model. Therefore, we estimate a c value between 0 and 2 cells/day.

This yields the model for our new healthy hepatocyte function and our new damaged hepatocyte function.

$$\frac{dH}{dt} = rH\left(1 - \frac{H+Z}{H_{max}}\right) - \eta H\left(1 - \frac{A}{A_{norm}}\right) - c\left|\frac{dO}{dt}\right|$$
$$\frac{dZ}{dt} = \eta H\left(1 - \frac{A}{A_{norm}}\right) - \delta_Z Z + c\left|\frac{dO}{dt}\right|.$$

Adding reperfusion damage, the model causes a large loss of cells immediately following treatment which means that at the time of treatment there is still a sizable amount of cell death that occurs. This additional cell death means that the danger threshold of liver damage before treatment is higher than the previously discussed 30%. It takes into account the possible damage from reperfusion and finds values where the liver would still maintain 30% of its healthy

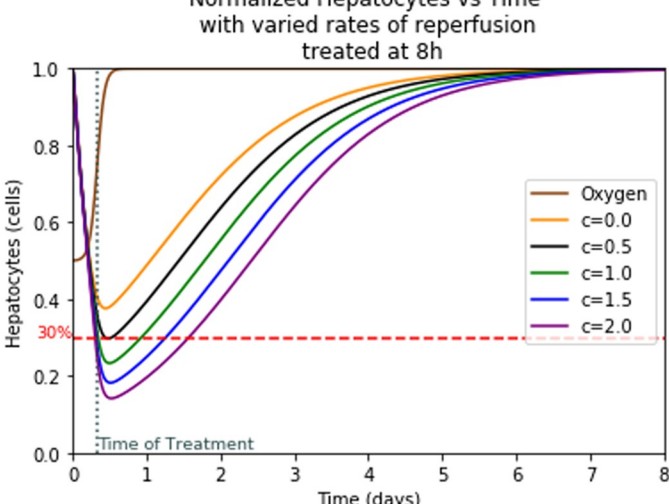

**Fig 10. Hepatocyte damage with varied rates of reperfusion treated at the 8th hour.**

hepatocytes after reperfusion. Fig 10 shows hepatocyte damage with varied rates of reperfusion treated at the 8th hour for different values of c which helps us estimate liver damage due to reperfusion better.

## 4 Conclusion

Ischaemic Hepatitis (IH) is a dangerous liver injury with a 50% mortality rate [5]. Cases of IH are caused by an underlying medical condition such as cardiac injury, respiratory injury, sepsis, or other types of shock. The underlying condition results in a decrease in the amount of oxygenated blood available to the liver, often greater decrease than can be mediated by the liver's natural hepatic arterial buffer response [1]. Without adequate oxygen, the liver cells (i.e., hepatocytes) cannot produce sufficient ATP, the cell's primary energy source. A reduction in ATP results in hepatocyte death through a process called necrosis. Extensive necrosis results in irreversible liver damage. The death of hepatocytes are identified by significant spikes in the blood serums such as AST, ALT, and LDH. Despite the severity of this condition, IH is limited to diagnosis via exclusion of other liver injuries and then treatment of the underlying cause. However, to best of our knowledge, there has not been any systematic dynamic study that attempts to understand the role of biomarkers and oxygen level on the concentration of healthy and damaged hepatocytes [25]. Hence, it is critical to develop an approach and a dynamic framework that may provide a reference for physicians to diagnose cases of IH and determine whether or not patients have reached critical damage based on their blood serum levels on real time basis.

In this study, we develop and analyze a novel mathematical model that capture a coupled dynamics of hepatocytes, cells death due to ischaemic hepatitis and relevant biomarkers (AST, ALT, and LDH). The derived model is based upon decay and regeneration of healthy and damaged hepatocytes in the liver. In order to model IH condition via dynamics of hepatocytes several assumptions are considered to understand the systematic damage of the hepatocytes. Due to the range of conditions that result in IH, we focus solely on cardiac injury. We assume that the cardiac injury decreases blood flow to the liver that results in a lack of oxygen. It is also assumed that any hepatocyte death is the result of necrosis due to insufficient ATP. This excludes hepatocyte death by any other processes. We consider that the hepatocytes produce

and consume ATP at the same rate. We also assume that any affect to ATP levels as a result of the LDH serum levels is minimal and does not effect the results of the model. The resulting model describes the death of the liver's hepatocytes, the reduction of ATP, and the resulting biomarker spikes due to a logarithmic oxygen function. This model is presented by a system of six differential equations. The biomarkers AST, ALT, and LDH are then compared with patient data from various studies [2, 27] and model results are validated. This comparison allows us to verify that the resulting biomarker levels are close to what is actually observed in cases of IH. Later model is modified to include additional hepatocyte damage due to the return of oxygenated blood to the liver in the process known as reperfusion. This component is added to help predict damage due to treatment and aid in the development of better treatment methods.

The model shows the effects of treatment time and oxygen level due to initial injury on the extent of hepatocyte death and the levels of the biomarkers. *We observe that a lower initial oxygen percent results in greater cell death, as is expected.* For example, it is seen in Fig 9a that with no treatment and 30% initial oxygen it takes approximately 12 hours for 70% of the hepatocytes to die. However, with 80% initial oxygen, it takes closer to 36 hours to reach the same percentage. This range shows that it takes between 12 and 36 hours for ischaemic liver injury to become irreversible. Similarly, *we observe that more cell death occurred with a delay in treatment, correlating with higher biomarker levels.* It is observed that LDH spikes the quickest and takes the longest to return to normal as compared to AST and ALT. AST spikes at similar rate to LDH but returns to normal much more quickly. The biomarker ALT spikes slower than the other two but returns to normal at a rate similar to that of LDH. As can be seen in Fig 9b, the biomarkers will return to normal despite a lack of treatment due to their natural flushing from the blood. In summary, *the initial level of oxygen and time to treatment since patient hospitalized is crucial to predict peak levels of biomarkers and hence, to estimate the level of liver damage.*

As seen in Fig 8a, the treatment time is a crucial factor in determining the extent of liver damage. By delaying treatment by just 4 hours the damage extends from 40% to over 60% for the most common initial oxygen. Given the initial oxygen percent in the body, *a physician can estimate the time since injury and diagnosis* by comparing the peak levels of patient's AST, ALT, and LDH levels upon diagnosis as displayed in Figs 5a, 6a and 7a. Similarly, if the approximate time of injury is known, comparing a patient's peak biomarker levels as in Figs 5b, 6b and 7b, physicians can estimate the percent oxygen present in body after damage. Thus the information of initial oxygen percent, combined with the knowledge of treatment time is useful to predict if the patient's liver damage has passed the critical 70% damage threshold for transplant, expediting the treatment process [26]. In the case, the threshold has likely been passed, the physician should start to look into the possibility of liver transplant and other forms of treatment.

In spite of some practical clinical results from this modeling study, there are some limitations. In our present model, we only consider Ischaemic Hepatitis due to *cardiac failure* [2, 12]. In future, we aim to consider IH due to other reasons. Thus, in future studies there may be the relevancy of incorporating other forms of cell deaths (such as apoptosis [47]) apart from necrosis. Moreover, such an extended focus model in future studies will be helpful to understand the impact of reperfusion in patients.

It would be ideal to find a rate that is slow enough to reduce reperfusion damage but quick enough to prevent irreversible damage due to ischaemia. Additional biomarkers can be added to our existing model in order to generate spectrum of clinical conditions observed in patients. This can include bilirubin which observes a moderate spike in cases of IH, or the blood's clotting factor. The clotting factor is related to the INR level, another common blood

measurement for which many of our sources contains data. In spite of these limitations, this study provides a first starting point where systematic mechanisms for IH can be identified and thoroughly evaluated. However, since new parameters are described here to study the dynamics of hepatocytes, there is strong need for relevant data, which is currently missing from empirical literature, to parameterize the model more rigorously.

## Supporting information

**S1 File.**
(PDF)

**S1 Data.**
(XLSX)

## Author Contributions

**Conceptualization:** Aditi Ghosh, William Lee, Anuj Mubayi.

**Formal analysis:** Claire Onsager, Andrew Mason.

**Investigation:** Claire Onsager, Andrew Mason, Anuj Mubayi.

**Methodology:** Claire Onsager, Anuj Mubayi.

**Project administration:** Anuj Mubayi.

**Resources:** Leon Arriola, William Lee.

**Software:** Andrew Mason, Leon Arriola.

**Supervision:** Aditi Ghosh, William Lee, Anuj Mubayi.

**Validation:** Aditi Ghosh, Claire Onsager, Andrew Mason.

**Visualization:** Anuj Mubayi.

**Writing – review & editing:** Aditi Ghosh, Anuj Mubayi.

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
