## [Decision Letter · Decision Letter 0]

16 Jul 2020

PONE-D-20-06445

The Role of Oxygen Intake and Liver Enzyme on The Dynamics of Damaged Hepatocytes: Implications to Ischaemic Liver Injury via A Mathematical Model

PLOS ONE

Dear Dr. Ghosh,

Thank you for submitting your manuscript to PLOS ONE. After careful consideration, we feel that it has merit but does not fully meet PLOS ONE’s publication criteria as it currently stands. Therefore, we invite you to submit a revised version of the manuscript that addresses the points raised during the review process.

We look forward to receiving your revised manuscript.

Kind regards,

Beicheng Sun

Academic Editor

PLOS ONE

Journal Requirements:

2. Thank you for including your funding statement; "none"

3. Thank you for including your competing interests statement; "No"

5. Please remove your figures from within your manuscript file, leaving only the individual TIFF/EPS image files, uploaded separately.  These will be automatically included in the reviewers’ PDF.

6. Please ensure that you refer to Figure 2 in your text as, if accepted, production will need this reference to link the reader to the figure.

Reviewers' comments:

Reviewer's Responses to Questions

**Comments to the Author**

1. Is the manuscript technically sound, and do the data support the conclusions?

Reviewer #1: Partly

2. Has the statistical analysis been performed appropriately and rigorously? 

Reviewer #1: Yes

3. Have the authors made all data underlying the findings in their manuscript fully available?

Reviewer #1: No

4. Is the manuscript presented in an intelligible fashion and written in standard English?

Reviewer #1: No

5. Review Comments to the Author

Reviewer #1: In this paper, the authors reported a mathematical model capturing dynamics of hepatocytes in the liver through the rise and fall of associated liver enzymes AST, ALT, and LDH related to condition of ischaemic hepatitis. The study appeared to be novel, and results were somewhat interesting. It is not clear, however, to this reviewer, based on reading this manuscript, how the clinical data was collected and to what extent the pathophysiological factors were taken into consideration during the construction of their models. Without a clear description of the data generation process, it is difficult to evaluate the validity of the mathematical models that the authors proposed in the study.

Specific comments are the following

1. The way that this paper was written is rather unusual, and I would suggest that the authors to make substantial revisions so that the format of the manuscript is consistent with others published in this journal.

2. The authors mentioned that they only considered one form of cell death, apoptosis, caused by multiple cytokines. The authors ignored an important pathological process, cytokine storm, which is a physiological reaction in humans and other animals in which the innate immune system causes an uncontrolled and excessive release of cytokines. Normally, cytokines are part of the body's immune response, but their sudden release in large quantities can cause multi-organ failure and eventually death. Because of cascade and positive feedback, many cytokines accumulate in damaged organs in a short time. So, without taking these factors into consideration, the significance of the model that the author constructed is rather questionable.

3. Although the authors also mentioned in the discussion that the consideration of apoptosis is too idealistic. Perhaps the inclusion of inflammatory cytokines such as IL-6 and TNF-α into the model would make it more convincing.

4. The sources of data were not described clearly. Critical information such as the number of patients, study locations, pathological degrees of the patients, as well as the demographic information of the patients (age, gender, etc) needs to be clearly provided, as they determine the validity of the models and many of them may be confounding factors for the calculations.

5. A better understanding and model of apoptosis will help improve our reperfusion model.

6. PLOS authors have the option to publish the peer review history of their article (what does this mean?). If published, this will include your full peer review and any attached files.

Reviewer #1: No

---

## [Author Response · Author response to Decision Letter 0]

24 Sep 2020

The mapped responses to the reviewer's comments can be found in the Response to letter file which has been attached.

---

## [Decision Letter · Decision Letter 1]

11 Dec 2020

PONE-D-20-06445R1

The Role of Oxygen Intake and Liver Enzyme on The Dynamics of Damaged Hepatocytes: Implications to Ischaemic Liver Injury via A Mathematical Model

PLOS ONE

Dear Dr. Ghosh,

Thank you for submitting your manuscript to PLOS ONE. After careful consideration, we feel that it has merit but does not fully meet PLOS ONE’s publication criteria as it currently stands. Therefore, we invite you to submit a revised version of the manuscript that addresses the points raised during the review process.

As you can see, the additional reviewer appreciated your work, but also suggested a couple of issues that need to be taken care of before an acceptance can be made.

We look forward to receiving your revised manuscript.

Kind regards,

Pavel Strnad

Academic Editor

PLOS ONE

Reviewers' comments:

Reviewer's Responses to Questions

**Comments to the Author**

1. If the authors have adequately addressed your comments raised in a previous round of review and you feel that this manuscript is now acceptable for publication, you may indicate that here to bypass the “Comments to the Author” section, enter your conflict of interest statement in the “Confidential to Editor” section, and submit your "Accept" recommendation.

Reviewer #2: (No Response)

2. Is the manuscript technically sound, and do the data support the conclusions?

Reviewer #2: Yes

3. Has the statistical analysis been performed appropriately and rigorously? 

Reviewer #2: Yes

4. Have the authors made all data underlying the findings in their manuscript fully available?

Reviewer #2: Yes

5. Is the manuscript presented in an intelligible fashion and written in standard English?

Reviewer #2: Yes

6. Review Comments to the Author

Reviewer #2: In this paper, authors presented a mathematical model to study the dynamics of ischaemic hepatitis at molecular level and calibrated the model with the published data available in literature. The stability analysis of equilibrium points was investigated. Numerical simulations were performed exhaustively. The paper is well written, clear and contains interesting results. There are minor comments in the attached review report to be addressed.

7. PLOS authors have the option to publish the peer review history of their article (what does this mean?). If published, this will include your full peer review and any attached files.

Reviewer #2: No

---

## [Author Response · Author response to Decision Letter 1]

19 Jan 2021

A response letter has been uploaded here.

---

## [Editor Report · Decision Letter 2]

21 Jan 2021

The Role of Oxygen Intake and Liver Enzyme on The Dynamics of Damaged Hepatocytes: Implications to Ischaemic Liver Injury via a Mathematical Model

PONE-D-20-06445R2

Dear Dr. Ghosh,

We’re pleased to inform you that your manuscript has been judged scientifically suitable for publication and will be formally accepted for publication once it meets all outstanding technical requirements.

Kind regards,

Pavel Strnad

Academic Editor

PLOS ONE
---

## [Editor Report · Acceptance letter]

15 Feb 2021

PONE-D-20-06445R2 

The Role of Oxygen Intake and Liver Enzyme on The Dynamics of Damaged Hepatocytes: Implications to Ischaemic Liver Injury via a Mathematical Model 

Dear Dr. Ghosh:

I'm pleased to inform you that your manuscript has been deemed suitable for publication in PLOS ONE. Congratulations! Your manuscript is now with our production department. 

Kind regards, 

on behalf of

Dr. Pavel Strnad 

Academic Editor

PLOS ONE